# Fear of the Queen's Speed: Trauma and Departure in *The Winter's Tale*

**Caroline Bicks**

Department of English, University of Maine, Orono, ME 04469, USA; caroline.bicks@maine.edu

**Abstract:** The essay applies trauma theory to early modern understandings of grief and its contagious after-effects to provide new ways to think about the figuring of trauma's reach into individual embodied minds and their environments, and about its larger impacts on narrative structures, theatrical spaces, and the people who populated them. To do so, I turn to Shakespeare's most deliberately tricky play, *The Winter's Tale*, and to its undeniably traumatized Queen Hermione, who defies the laws of time, space, and motion to an extent unmatched by any other human character in his canon. The essay explores how Shakespeare imagines and mobilizes the aggrieved Hermione; and how her departure and repeated, belated returns play out different forms and effects of traumatic response. These include the gaps and eruptions endemic to the processes of accommodating the impossible and listening to stories that are structured around absence and aporia. It is my contention that in his later play he was experimenting with how those effects could spill out beyond the brains and bodies of trauma's original victims, and transform the people and spaces beyond them.

**Keywords:** trauma; early modern passions; Shakespeare; grief

> Grief fills the room up of my absent child,
> Lies in his bed, walks up and down with me,
> Puts on his pretty looks, repeats his words,
> Remembers me of all his gracious parts,
> Stuffs out his vacant garments with his form.
>
> —Shakespeare, *King John* (3.4.93-97)[1]

The turn in recent decades toward studies of embodiment, cognition, and the passions has produced a wealth of scholarship on how early moderns imagined the relationship between a person's physical, mental, spiritual, and emotional experiences. Some critics use the term *body-mind* to capture one popular view (based in Aristotelian ideas of the embodied soul and Galenic humoral theory) that these were interdependent phenomena; and that the reciprocal dynamics between one's brain, body, and emotions could affect and be affected by the people and environments around them.[2]

With these frameworks in place, we can better appreciate why it is that when *King John*'s Constance enters the French camp lamenting the loss of her imprisoned child, Arthur, she expresses her grief as a series of material afflictions—its presence painful and contagious: "too well, too well I feel / The different plague of each calamity" (3.4.59-60). Fearing for her son's imminent death, she vividly describes how grief spreads and circulates between spaces and bodies, living and dead: it already "fills the room up" in Arthur's absence, and stuffs itself into his bed and clothes, impersonating his expressions and never leaving her side. Although Constance enters the scene tearing her hair out, she has not lost her mind yet—much to her dismay: "I am not mad; I would to God I were, / For then 'tis like I should forget myself. / O, if I could, what grief should I forget!" (3.4.48-50).

Shakespeare's plays are rife with characters who die of grief.[3] And, as Erin Sullivan's work with seventeenth-century London Bills of Mortality demonstrates, real early modern English people were officially doing so as well.[4] Constance's specific figuring of grief's

calamitous arrival as a plague, then, would have registered as more than just a metaphor to the play's audiences: both were unpredictable repeat offenders in early modern England, wreaking havoc on its population.[5] Fittingly, physician Stephen Bradwell included grief's ruinous effects in his treatise on the plague. He writes that extreme sorrow "afflicts the Heart, disturbes the Faculties, melts the Braine, vitiates the humours, and so weakens all the principall parts; yea, sometimes sinkes the Body into the grave." Fear, which Bradwell describes as grief's close cousin, was "the most pestilently pernicious" of the passions: it "gathers the heat and Spirits to the heart, and dissolves the Brayne, making the moysture thereof shed and slide downe into the externall parts, causing a chilnesse and shaking over all the Body . . . It brings a lethargie upon the Organs of motion, and condemnes the heart to deadly suffrings."[6]

The painful, possibly fatal effects of grief and fear undoubtedly seemed real to early modern subjects, and are legible to us thanks to recent work on embodied cognition and the passions. But listening now to Constance's wish to lose her mind and forget her grief, when we are in the midst of a global pandemic that has triggered a widespread mental health crisis, it is impossible not to hear another register to her lament: trauma. Cathy Caruth, whose work is a touchstone for scholars of early modern emotions, describes how trauma is experienced "as a temporal delay that carries the individual beyond the shock of the first moment," a phenomenon that marks his or her cognitive departure from the site of the traumatic experience. As a result, the event is not "assimilated or experienced fully at the time, but only belatedly, in its repeated *possession* of the one who experiences it." The forms that a traumatic response can take include "repeated, intrusive hallucinations, dreams, thoughts or behaviors stemming from the event, along with numbing that may have begun during or after the experience."[7]

Trauma theory helps us hear in Constance's lament a desire for self-preservation, not suicide. In seeking to detach from her memory, she expresses a wish to depart from the site of her trauma—to numb herself from re-experiencing the pain of her son's loss and of feeling grief's expansive and over-stimulating intrusions. In so doing, she envisions her body and mind as separable entities, and so reveals an alternative to the unified early modern body-mind and its operations that have dominated recent scholarship.[8]

Constance wants to lose her mind because it is her continued attachment to her cognitive faculties—not just memory, but also reason—that she claims will lead her to kill herself:

> Preach some philosophy to make me mad,
> And thou shalt be canonized, Cardinal;
> For, being not mad, but sensible of grief,
> My reasonable part produces reason
> How I may be delivered of these woes,
> And teaches me to kill or hang myself. (3.4.51-56)

She identifies this same "reasonable part" as the enemy that works against her best interests by tempting her to love grief for its impersonations of her absent son. After she imagines it walking and talking like Arthur, and following her every step—like the "repeated, intrusive hallucinations" Caruth identifies as endemic to a traumatic response—she declares: "Then have I *reason* to be fond of grief" (3.4.98, emphasis mine). Constance clearly articulates a desire to not follow her grief-induced reasoning, and to not be "sensible of grief" at all—to resist its threats to possess her thoughts, dreams, and affections. We later hear that Constance may have gotten her wish: rumor has it that she was "in a frenzy" when she died, free of remembering grief and so, perhaps, of feeling the painful, contagious effects of each calamity (4.2.122).[9]

The past twenty years have witnessed an increasing surge in the application of trauma theory to early modern studies. Literary critics are especially drawn to Caruth's description of how trauma disrupts narrative coherence and challenges conventional understandings of how we experience and remember history. Because of the initial cognitive departure of

the individual from the site of his or her trauma, its history is structured around aporia, the "impossibility of knowing that first constituted it." The history that a traumatized individual must carry, then, is also impossible: it can never be fully grasped, known, articulated, and situated in the past.[10]

Applications of trauma theory to earlier time periods, however, have raised concerns about using a modern-day lens to read the past. In introducing their recent collection, Erin Peters and Cynthia Richards explain that the history of trauma theory (which originated in nineteenth- century psychoanalysis and developed within twentieth-century scholarship on war and PTSD) has reinforced the notion that it is "a thoroughly modern experience." While we must recognize specific historical and cultural frameworks when applying it to pre-modern eras, they argue, we should not shy away from the project for fear of "accusations of anachronism or retrospective diagnosis."[11] As Judith Pollmann reminds us, "[t]he twentieth century did not invent victimhood." She asserts that the reason "we find so little trace of post-traumatic stress disorder in early modern sources may be because early modern culture struggled to articulate such pain."[12] And yet, as Constance's gripping and specific expressions demonstrate, these traces not only existed but—in the case of early modern drama—commanded the stage, filling up the room (or the outdoor playing space) like grief itself.

My goal in this essay is to consider how this understanding of grief and its contagious after-effects (evident across multiple genres and discourses) can provide new ways to think about the figuring of trauma's reach into individual bodies, minds, and their environments, and about its larger impacts on narrative structures, theatrical spaces, and the people who populated them. To do so, I am turning to Shakespeare's most deliberately tricky play, *The Winter's Tale*, and to its undeniably traumatized Queen Hermione, who defies the laws of time, space, and motion to an extent unmatched by any other human character in his canon. Accused of adultery and treason by her paranoid husband, King Leontes, the pregnant Hermione is banished from her young son, Mamillius; forced to give premature birth in jail to a daughter whom Leontes then throws out of his kingdom; hustled out in public to endure a trial before her postpartum body has had time to heal; and suffers an allegedly fatal collapse in reaction to her son's sudden death. The drama is riddled with inconsistencies, the most puzzling of which collect around the series of encounters characters have with Hermione after her presumed death: Leontes claims that he saw his wife's dead body; and she appears as a ghost to Antigonus, the man charged with exposing her newborn. Both events are impossible given the play's final surprise reveal that Hermione never actually died, but has been hiding away for sixteen years.

Scholars have produced a variety of illuminating studies exploring the typological, mythical, cultural, and religious associations that Hermione evokes—first as a pregnant woman and mother, then as a ghost, and finally as a statue that seems to come to life.[13] That said, Hermione is so weighed down with symbolism and literary predecessors—the sacrificial mother, the forgiving wife, Alcestis, Griselde, Pygmalion's statue, the Virgin Mary—that we risk burying the living, feeling person that Shakespeare imagined beneath it all. Earlier studies that did address Hermione's human embodied state (including my own) focused almost exclusively on her maternity, and on the effects of her pregnant and postpartum conditions on the play's themes and characters.[14] Her reproductive body—and its bookend, the aged, post-menopausal one—continue to animate more recent studies as well, including those informed by the newer approaches to embodiment and the passions that have enriched our understanding of early moderns' lived experiences.[15]

It is high time that we update our view of Hermione and leverage the interpretive tools we now have, including those being forged by early modern trauma studies, to recognize the dynamic and expansive operations of her body-mind, but also to acknowledge that the early modern mind might detach from the body as a form of self-protection; to listen to how she describes her emotions, and to track the impact of her traumatic responses on herself and her environments. As Constance's vivid descriptions of grief demonstrate, Shakespeare understood these complex and, at times, inconsistent dynamics, infusing them

into his characters' figures of speech. It is my contention that in his later play he was experimenting with how trauma's effects could spill out beyond its original victims and transform the people and spaces beyond them.

Furthermore, our understanding of *The Winter's Tale*'s thematic and structural reliance on the image of tale-telling can be greatly enriched by considering it alongside trauma theory's emphasis on the importance of and barriers to therapeutic listening. As Caruth writes, "[t]he history of a trauma, in its inherent belatedness, can only take place through the listening of another." But this is a challenging role to take on: "To listen to the crisis of a trauma . . . is not only to listen for the event, but to hear in the testimony the survivor's departure from it; the challenge of the therapeutic listener, in other words, is *how to listen to departure*."[16]

In what follows, I explore how Shakespeare imagines and mobilizes the aggrieved Hermione; and how her departure and repeated, belated returns play out different forms and effects of traumatic response. These include the gaps and eruptions endemic to the processes of accommodating the impossible and listening to stories that are structured around absence and aporia. I am inspired here by one of the first applications of trauma theory to Shakespeare, Heather Hirschfeld's 2003 article on *Hamlet*'s incessant repetition, "writ large across the play," which she argued is symptomatic of "the traumatic compulsion to return to a moment that . . . must be but cannot be grasped." These repetitions, refracted through the play's mirrored plotlines, suggest that "even the drama itself cannot fully process Hamlet's situation."[17] As we shall see, the same can be said of how *The Winter's Tale* fails at key moments to process Hermione's grief and its after-effects.[18] These failures, I would argue, mark Shakespeare's inventive refigurings of trauma. When we look and listen for them, we can recognize what grief theorist Lesel Dawson describes as "the role psychic distress plays in creativity," and appreciate its reciprocal effects: "art's ability to reconfigure the imagination."[19]

## 1. "I have / That honourable grief lodged here which burns"

Critics of *The Winter's Tale* have long been drawn to the etiology of Leontes' paranoid jealousy and quick derangement.[20] More recent studies of embodied cognition continue to favor Leontes' felt experiences, to the near total exclusion of Hermione's or anyone else's.[21] In the same vein, when critics apply trauma theory to the play, they do so primarily to analyze Leontes' psychic struggles.[22] While there is obvious value to exploring his interconnected physical, mental, and spiritual torments, this scholarly trend has resulted in another kind of imbalance: his emotions and trauma far outweigh Hermione's. Typic-ally she is relegated to the role of healer, saving her husband and his disrupted psyche via her cathartic reanimation and return at the end.

A further challenge to making her emotions central to a reading of the play is that they are primarily filtered through Leontes' biased interpretations of them—and then subsumed within his own psychophysiological responses. Leontes reads Hermione's very pregnant body as "[t]oo hot, too hot" while he watches her with his best friend, Polixenes (1.2.100). He connects her heat (which, of course, he cannot reliably assess) to what he imagines is her sexual excitement. The sight spreads through his brain and body to cause "*tremor cordis*": "my heart dances, / But not for joy; not joy" (1.2.112-113). And this is just the start of what will become pages of his self-described torments.

Given the poetic force and prolixity of his expressions, one can see why scholars are drawn to Leontes' explicitly embodied passions. Hermione speaks of hers in a far more modest and muted way in this important opening scene between them—an effect, no doubt, of the public context in which she expresses them (unlike him she has no asides or soliloquies), and her adherence, as the King's wife, to courtly protocol. She tells Leontes, "I love thee not a jar o'th' clock behind / What lady she her lord," and that she "long[s]" for his praise (1.2.43-44,103).

It isn't until her next scene, when she is alone with her young son, Mamillius, and her female attendants that she expresses any negative feeling, telling her ladies to take her son

from her because he "so troubles me / 'Tis past enduring" (2.1.1-2). She does not say that the physical challenges of an advanced pregnancy are affecting her emotional endurance here, although one lady does comment on how she "rounds apace" (2.1.17). In any case, her annoyance is short-lived and seemingly mild judging by her request twenty lines later that Mamillius come sit by her and whisper his tale in her ear (2.1.34).

The first time she speaks of her emotions as an explicitly embodied experience is after Leontes barges in on this scene, accuses her of fathering Polixenes' child, takes her son away from her, and orders that she be imprisoned. She has had no time to prepare for these shocking events, a condition that early modern writers on the passions singled out as especially dangerous.[23] Nonetheless, she articulates her emotions in a composed manner, directing her speech to the lords who have accompanied Leontes to do his cruel bidding:

> I am not prone to weeping, as our sex
> Commonly are; the want of which vain dew
> Perchance shall dry your pities. But I have
> That honourable grief lodged here which burns
> Worse than tears drown. (2.1.110-114)

Visible expressions of sorrow are, in her view, vain, although she recognizes the power of womanly tears to move an audience to pity. And yet, perhaps knowing that she will be "measure[d]" by these men (2.1.116), and caring for the perception of her honor above all else, she does choose to reveal something of her embodied emotional state to them: her "grief"—"honourable" like her, but painful nonetheless.

Hermione's articulation of her burning grief signals an engagement with multiple early modern discourses—not just medical ones, but also the moral and religious doctrines that inform her claim that she must be "patient till the heavens look / With an aspect more favourable" (2.1.108-109). In her work on the spiritual dimensions of affect in the writings of Thomas Wright (whose early modern text on the passions of the mind is a touchstone for emotion studies), Erin Sullivan explores "the complex and at times unstable way in which passion was bound up in questions of mind, body and soul in the period, if not always holistically unifying their processes then certainly constellating them."[24] She joins other critics in pushing back against the notion that all early modern discourses constructed these phenomena as entirely interdependent. Religious belief systems in particular challenged a monist view of the body-mind based on humoral theory.

The complexity of these dynamics is evident in the early modern treatise of theologian Edward Reynolds, in which he describes the importance of sharing grief rather than allowing it to burn unvented—a condition that poses real threats to body, mind, and soul, even as it allows for generative sympathy:

> [I]n matter of Griefe, the *Mind* doth receive (as it were) some lightnesse and comfort, when it finds it selfe *generative* unto others, and produces *sympathie* in them: For hereby it is (as it were) disburthened, and cannot but find that easier, to the sustaining whereof, it hath the assistance of anothers shoulders . . . That *Griefe* commonly is the most *heavie,* which hath fewest *vents,* by which to *diffuse* it selfe: which, I take it, will be one occasion of the *heavinesse* of *infernall torment*.[25]

Since she is not "prone to weeping," Hermione instead may be seeking to produce the sympathetic response Reynolds describes by speaking of her burning grief—a decision that also may help diffuse some of its "*infernall torment*." Although no stage direction specifies where it is Hermione points when she claims her grief is "lodged here," writers of all backgrounds and genres agreed that, in Wright's words, "the heart is the place where the passions allodge."[26] A gesture toward the heart at this moment would have been a powerful piece of emotional theater, localizing and intensifying the combined physical, mental, and spiritual nature of her grief for her audience.

Hermione follows this first naming of her embodied emotional pain with her first explicit reference to her pregnancy as a physical condition requiring, in Reynolds' words, "anothers shoulders." This is a situation that she tells her husband he cannot choose but

recognize and accommodate: "Who is't that goes with me? Beseech your highness / My women may be with me, for you see / My plight requires it" (2.1.118-120). While she undoubtedly is referencing her endangered physical state here, *plight* carried two meanings at the time that speak to early modern views of embodiment, cognition, and the passions: it could reference a bodily condition and a state of mind. And, at this point, Hermione directs her audience's attention to both.

If she has achieved any lightness of mind by revealing her grief to what she hopes are sympathetic listeners, such relief proves short-lived, for her body soon goes into early labor. When Paulina comes to the jail and asks the midwife how Hermione fares, the latter replies: "As well as one so great and so forlorn / May hold together. On her frights and griefs, / Which never tender lady hath born greater, / She is, something before her time, delivered" (2.2.25-28). Catalyzed by fear, the "most pestilently pernicious" of the passions (to recall Bradwell), Hermione's grief appears to have spread well beyond the lodging of her heart to affect the natural course of her pregnancy. Although this is the midwife's assessment and not Hermione's, untimely births were attributed to extreme emotions.[27] By featuring them here, Shakespeare calls attention to their perceived harmful effects on the combined body-mind.

The description of Hermione's limited ability to "hold together" under this duress, however, is also suggestive of traumatic experience, and of the unique role that the mind plays in negotiating it. Zackariah Long connects the early modern notion of *uncollectedness* to the Freudian concept of *unbinding*, in which the pysche cannot master all of the stimuli presented and has no time to prepare the ego for such disruptions.[28] If earlier she was able to remain composed in the face of Leontes' surprise disruption of her domestic space and his abrupt removal of her son, she is less able to do so in prison as she suffers the untimely exit of her daughter from her body.

When Hermione does start speaking for herself again in the ensuing trial scene, however, she does so with characteristic restraint and control. While she points to the ways in which her postpartum body is suffering at Leontes' tyrannous demands that she come out in public before she has "got strength of limit," she first speaks to her grievous emotional state (3.2.104). Her unhappiness is measureless, but her words contain it within an eloquent rhetorical comparison:

> You, my lord, best know—
> Who least will seem to do so—my past life
> Hath been as continent, as chaste, as true,
> As I am now unhappy; which is more
> Than history can pattern, though devised
> And played to take spectators. (3.2.30-35)

At this point in the play and in Hermione's experience, Leontes is the sole violator of history and memory: he is the only one who willfully denies the honor and chastity of her "past life," which she claims he should "best know." But with her suggestion that her unhappiness cannot fit within the bounds of any patterned history (including a drama "played to take spectators") she signals what will become a recurring feature of the drama's second half: her embodied emotions and their traumatic after-effects (which are, at times, disembodied) will return in a series of reconfigured forms, disrupting all attempts to contain her history and to relegate trauma to the past.

For now, however, Hermione takes command of fear and grief by folding them into her claim that she welcomes death, since Leontes has taken away all her worldly joys—his favor and her children: "For life, I prize it / As I weigh grief, which I would spare," or dispense with (3.2.40-41). The threat of death, "The bug you would fright me with," she tells Leontes, "I seek" (3.2.90-91). Her refusal to succumb to these passions casts an unflattering light on Leontes' own conspicuous (and seemingly manipulative) display of his embodied emotional suffering as he opens his tyrannous proceedings: "This sessions, to our great grief we pronounce, / Even pushes 'gainst our heart" (3.2.1-2).[29]

Hermione claims that the only thing keeping her alive is "mine honour, / Which I would free," an event that she believes will happen once the oracle, to which Leontes has sent to confirm her guilt, is read out loud (3.2.108-109). Her joyful response to its confirmation of her innocence—"Praised!"—is followed, however, by grievous and disabling news: a servant enters to tell Leontes that "your son, with mere conceit and fear / Of the Queen's speed, is gone," and Hermione falls to the ground (3.2.142-143). What Leontes diagnoses as an "o'ercharged" heart from which his wife will "recover," instead proves, according to Paulina, to be "what death is doing" (3.2.147-148). Hermione is carried off stage, and Paulina reenters to swear that she has perished. Leontes laments his error, and asks her to "bring me / To the dead bodies of my queen and son" (3.2.233-234).

As Kaara Peterson's work on early modern hysterical illness demonstrates, audiences likely were acquainted with the medical phenomenon of women suffering from uterine suffocation who appear to die, only to be revivified after three days. She argues that "Hermione's tragic postpartum death, 16-year absence, and reanimation pointedly mirror the hibernating hysteric," a theory that offers a reasonable solution to how it is that Leontes (and perhaps Paulina) could see Hermione's dead body when she was never actually dead.[30]

But this focus on Hermione's postpartum uterine condition risks reducing her experiences down to the sum of one part. It does not take the material effects of sudden emotional shock into account. Hermione experiences the near simultaneous effects of joy and grief, hearing the oracle and then the news of her son's death within a few lines. Such "Contraritie of Passions," according to Thomas Wright, tosses one's heart "like the Sea with contrary winds." He goes on to describe "[a]n other Disquietnesse . . . , which to many happeneth, and that welnie upon a sodayne: For some times a man will bee in the prime of his joy, and presently a sea of grief overwhelmeth him."[31] Wright's words echo the depiction of Bellaria's emotional state before she dies in Robert Greene's *Pandosto*, Shakespeare's immediate source for *The Winter's Tale*:

> [T]here was worde brought him that his young sonne *Garinter* was sodainly dead, which newes so soone as *Bellaria* heard, surcharged before whith extreame joy, and now suppressed with heavie sorrowe, her vitall spirites were so stopped, that she fell downe presently dead, & could be never revived.[32]

Furthermore, we miss another key register to Hermione's experience when we attribute her demise to a uterine disorder: Shakespeare expands on Greene's narrative here by describing the shared trauma between mother and son that initiates her collapse. Mamillius appears to have suffered a fatal traumatic response triggered by two embodied conditions: his shocking separation from his mother, from whom he is "barred, like one infectious," in Hermione's words (3.2.96); and his "fear" of her "speed" (or what will become of her), in the words of the servant.

It is not his mother, of course, but that "most pestilently pernicious" passion—fear—that seems to be the real contagion. Its traumatic after-effects spread to Hermione and sink them both into the grave. The servant's turn of phrase here further connects Mamillius's passions to his mother's by combining grief's contagious spread with the trauma victim's sudden cognitive departure: Mamillius's "fear / Of the Queen's speed" appears to have powered and accelerated her exit.

## 2. "with shrieks / She melted into air"

Unlike her prototype in Greene, Hermione will come back. But what happens to her aggrieved and traumatized body, mind, and passions once they no longer fit into the pattern of Greene's history, or into the bounds of narrative logic more broadly? Even if we consider that Hermione may have suffered from the suffocation of the womb, or the shock of contrary emotions, no early modern medical theory can make sense of her appearance in the next scene as a ghost who visits Antigonus on board his ship. Nor does her spectral return while still alive adhere to any rules of Christian doctrine. In addition to the physical, spiritual, and formal problems this episode presents, it muddles a coherent reading of Hermione's character as modest and benevolent. Perhaps this is why few scholars attend

to this riddling return, relegating it to a dream when they do.[33] Since Shakespeare does not stage the episode for his audience, this is a theory that is easy to let go unchallenged.

But when we fail to recognize Hermione's spectral presence, we fail the larger challenge of the therapeutic listener, to recall Caruth's description of it: "*to listen to departure.*" It is imperative that we situate this impossible event, then, within the theoretical and lived frameworks of traumatic experience, and that we listen to Hermione's ghost. Antigonus relates the story of her visitation once he has landed on the shores of Bohemia.[34] His only on-stage audience is the baby whose miserable fate, death by exposure, he has been ordered by Leontes to seal:

> Come, poor babe.
> I have heard, but not believed, the spirits o' the dead
> May walk again. If such thing be, thy mother
> Appeared to me last night, for ne'er was dream
> So like a waking. To me comes a creature,
> Sometimes her head on one side, some another;
> I never saw a vessel of like sorrow,
> So filled and so becoming. In pure white robes
> Like very sanctity, she did approach
> My cabin where I lay, thrice bowed before me,
> And, gasping to begin some speech, her eyes
> Became two spouts. The fury spent, anon
> Did this break from her: 'Good Antigonus,
> Since fate, against thy better disposition,
> Hath made thy person for the thrower-out
> Of my poor babe, according to thine oath,
> Places remote enough are in Bohemia.
> There weep, and leave it crying; and for the babe
> Is counted lost for ever, Perdita
> I prithee call't. For this ungentle business
> Put on thee by my lord, thou ne'er shalt see
> Thy wife Paulina more.' And so with shrieks
> She melted into air. Affrighted much,
> I did in time collect myself, and thought
> This was so, and no slumber. (3.3.14-38)

Antigonus initially displays skepticism, typical of early modern Protestant doctrine, regarding the existence of ghosts: "I have heard, but not believed, the spirits o'th'dead / May walk again." But he follows this up by allowing for their possible reality: "If such thing be, thy mother / Appeared to me last night." And although he is confused at first as to whether or not he was asleep ("for ne'er was dream so like a waking"), he declares—after he takes time to "collect myself"—that he "thought / This was so, and no slumber." He expresses clear markers of cognitive collectedness here. We would be remiss to ignore them and to relegate Hermione's ghost unequivocally to his dream-world.

But how are we to grasp the impossibility of her simultaneous existence as a ghost and a living being? Cathy Caruth's account of trauma's origin in the Greek word for *wound* offers some provocative strategies:

> [T]he wound of the mind—the breach in the mind's experience of time, self, and world—is not, like the wound of the body, a simple and healable event . . . [It] is experienced too soon, too unexpectedly, to be fully known and is therefore not available to consciousness until it imposes itself again, repeatedly, in the nightmares and repetitive actions of the survivor . . . [T]rauma seems to be much more than a pathology or the simple illness of a wounded psyche: *it is always the story of a wound that cries out*, that addresses us in the attempt to tell us of a reality or truth that is not otherwise available.[35]

Hermione's spectral, sometimes shrieking intrusion into the play embodies this definition of trauma as "the story of a wound that cries out" in an attempt to tell us "a reality or truth that is not otherwise available." This is not to detract from her status as an actual victim of trauma in the play, but rather to expand our view of traumatic experience's after-effects to include their potential to disrupt and reimagine old forms when mobilized within fictional modes. Such a reading demands that we suspend some of our habits of critical disbelief—that we, like Caruth's therapeutic listener, let go of what we think we know and the tools we think we have for knowing it: "By carrying that impossibility of knowing out of the empirical event itself, trauma opens up and challenges us to a new kind of listening, the witnessing, precisely, *of impossibility*."[36]

This approach opens up exciting new interpretive avenues. We might, for example, consider that Shakespeare created Hermione's impossible spectral appearance in order to conceptualize trauma's temporal delay—to reimagine it as a protective time-traveling act that allows her to depart from the scene of her traumatic experience and arrive simultaneously somewhere else. Or that he was exploring what it might look like to dramatize the mind's spectacular, self-preserving split from its pain-ridden, embodied form; or to imagine an unleashed traumatic fury that does not end, as it allegedly does for Constance, in a self-imploding frenzy—but rather shrieks and commands. We also might discover more nuance to why it is he does not stage the episode, but rather selects only Antigonus to bear witness and listen to Hermione's trauma. Caruth writes that "one challenge of this listening is that it may no longer be simply a choice: to be able to listen to the impossible, that is, is also to have been *chosen* by it, *before* the possibility of mastering it with knowledge." Referencing psychiatrist Lenore Terr, she continues: "This is its danger, as some have put it, of the trauma's 'contagion,' of the traumatization of the ones who listen . . . . But it is also its only possibility of transmission."[37]

From this perspective, we can imagine Shakespeare's choices here as part of a larger strategy for dramatizing the obstacles to trauma's transmission beyond the self, and the perceived dangers to those tasked with receiving it. Antigonus is chosen to witness the impossibility of Hermione's ghost, but he attempts to pattern her shifting forms according to the history he knows, or thinks he knows. He describes her first incarnation, silent and dressed in "pure white robes," as reflecting the essence of her modesty and goodness: "in very sanctity she did approach." He describes her grief, which he interprets from the motions of her head, as similarly contained by her saintly form: "I never saw a vessel of like sorrow." But then her passions explode, shattering that vessel as she "gasps to begin some speech," and her eyes turn into spouts until her "fury" is spent. Her words finally "break from her" in the wake of this passionate overflow.

When she speaks, Hermione's ghost returns to the site of one of her wounds, the throwing out of her baby (and Antigonus's role in it). The loss of her daughter is a traumatic event that she names twice and in two forms ("lost" and "Perdita"), but it is not the one Shakespeare dramatizes for his audiences as the most cataclysmic. That, of course, would be Mamillius's death, an episode that Antigonus did not witness, since he left Sicilia before it occurred; and one that Hermione's ghost cannot render in any intelligible form here. This particular narrative rupture poignantly figures the unknowability of traumatic experience and the impossibility of its account—the story of a wound that cries out, but cannot be grasped. A shriek that melts into air.

Antigonus is "affrighted" by what he has witnessed, but he is unable to catch the ghost's passions. She commands him to absorb and circulate the weeping that begins with her grief and will continue with her daughter's tears: "weep, and leave it crying." But as he lays the baby down, he claims that "Weep I cannot" (3.3.50). Ultimately, he has not listened to her trauma in the way Caruth describes. He also fails to hear the history of her honor—the one personal legacy that she explicitly had claimed was the one she cared more for than life itself (and obviously a different kind of history from the impossible traumatic one that the scene cannot accommodate). Instead, he interprets her directive as evidence

that Bohemia's king, Polixenes, is the father: "Poor wretch, / That for thy mother's fault art thus exposed / To loss and what may follow" (3.3.48-50).

Seven lines later, Antigonus will be, to quote his final words, "gone for ever!" — ripped apart limb by limb (3.3.57). This event, like the initial experience of trauma, is one to which we do not have access, although Shakespeare imagines (and the published play-text famously records in a stage direction) the terrifying threat of its return: "*Exit, pursued by a bear.*" Despite his poor listening, perhaps Antigonus has caught trauma's contagion after all and not been able to "hold together"—a cluster of embodied psychic metaphors that Shakespeare mixes here to invent a gruesome new one for trauma's dis(re)membering effects on the body and mind, one that he then sets loose upon the stage and into the play-world.

### 3. "attentiveness wounded his daughter"

While all of this unbounded trauma has been shrieking and tearing bodies apart in Bohemia and its watery environs, Leontes has been back in frozen Sicilia, attempting to divide and contain what Hermione and Mamillius's conjoined passions have unleashed. He consigns their bodies to one space, and organizes the emotional fallout according to his own disciplined expressions of feeling:

> One grave shall be for both. Upon them shall
> The causes of their death appear, unto
> Our shame perpetual. Once a day I'll visit
> The chapel where they lie, and tears shed there
> Shall be my recreation. (3.2.234-238)

In Leontes' vision, traumatic effects are reduced to known and recorded causes; shame is stabilized in perpetuity; and grief takes visible and controllable shape, first as the bodies of his dead wife and child (he ends by instructing Paulina to "Come, and lead me / To these sorrows") and finally as a grave within a chapel to be visited, wept over, and departed from at regularly appointed hours. But the play soon fast forwards sixteen years, sliding over a "wide gap" of time (as Time himself tells us when he enters after Antigonus' death to hurry along the shocking transition from tragedy to comedy [4.1.7]). This temporal gap and generic mismatch are obvious metaphors for trauma's disruption of narrative consistency and of any reliable, chronological understanding of history and personal experience.[38]

It is beyond the scope of this essay to examine all of the temporal-spatial disruptions, belated returns, and interrupted narratives that riddle the play's remaining acts. But I will finish by focusing on two key examples that center on Hermione, and that best exemplify trauma's contagious, unbounded features—as well as its creative possibilities. This means that I will not be analyzing Leontes' trauma in his exchange with Paulina, poignantly expressed as it is, where he begs her not to speak of Mamillius, for "Thou know'st / He dies to me again when talked of" (5.1.118-119); nor will I focus on Paulina's powerful description of Hermione's ghost returning to haunt him should he marry again.

Instead, I want to look at a small moment within the extended episode that is narrated by a series of gentlemen in the play's penultimate scene. Together, they "make a broken delivery," taking turns recounting pieces of the multiple reunions happening off-stage between (most) of the play's surviving main characters (5.2.8). All of them are described by the gentlemen as experiencing the extremes of joy and sorrow that early moderns understood as a potentially dangerous collision of emotional contrarieties. But the specific act of traumatic transmission occurs when Leontes tells Perdita the history of her mother's death. Here is how the Third Gentleman describes it:

> One of the prettiest touches of all, and that which angled for mine eyes—caught
> the water, though not the fish—was when at the relation of the Queen's death,
> with the manner how she came to't bravely confessed and lamented by the King,
> how attentiveness wounded his daughter till from one sign of dolour to another
> she did, with an 'Alas', I would fain say bleed tears; for I am sure my heart wept

blood. Who was most marble there changed colour. Some swooned, all sorrowed.
If all the world could have seen't, the woe had been universal. (5.2.74-83)

Perdita's attentiveness, her intense listening, is what "wound[s]" her. This is the only appearance of the word—a translation of *trauma* itself—in the play; and it occurs, fittingly enough, at this crucial moment when she is doing the kind of listening to trauma that Caruth defines as necessary for it to be heard and transmitted.

The wound figured here signals Perdita's absorption of the story of her mother's death. The tale moves into her attentive ears and through her body, showing itself in many signs of "dolour," including her verbal expression of grief ('Alas') and her bleeding tears. The gentleman's own eyes water as he witnesses her grief. What he relates is an account of trauma's contagious pain, but also, as theorist Roger Luckhurst argues, "the possibility of [its] release into narrative"; and, finally, its particular power, once released, to unite an audience in their sorrow and to bring even those who were "most marble" back to life to join in the "universal woe."[39]

But Hermione is not the one telling her story here. Her history—the speed with which she absorbed her son's trauma and departed—is subsumed by Leontes' need to "bravely" confess his responsibility for "the manner how she came to't." In his version, she long ago arrived at her final resting place, and he was the one who drove her "to't." Furthermore, we, the audience, are not the ones in the room listening. There is something inherently frustrating and incomplete, then, about the gentleman's description of therapeutic release. "*If* all the world could have seen't, the woe had been universal"—but few people, including us and, most importantly, Hermione, are granted access. She is still "most marble," and waiting in the wings.

There is a long critical tradition of reading Hermione's transition from apparent statue to living woman in the final scene as a commentary on early modern theater's power over its audiences.[40] Rather than focus on her embodiment of art, however, I want to end where this essay began: by listening to what she says about her specific experience; asking what it might tell us about early modern understandings of trauma; and appreciating how Shakespeare was reimagining its transformative effects on individual bodies and minds, and the people and spaces beyond them.

Like her ghost, Hermione-as-statue seems to explode all rules of time, space, and motion when she descends from her pedestal and begins to move and speak. But, unlike that furious, shrieking specter, she is visible and audible to everyone in the theatrical space—the on-stage audience and those who have come to see Shakespeare's play. No one can turn a deaf ear to her, at least at first. Hermione's belated return at the end of the play seems to enact a kinder, gentler transmission of trauma and its effects. This is a wound that does not cry out, but still (to return to Caruth) "addresses us in the attempt to tell us of a reality or truth that is not otherwise available."

An oft-noted curiosity of the ending, however, is that the main characters, so eager to experience Hermione's return, are so quick to cut the testimony of her survival short once she begins it.[41] Paulina will interrupt a potential exchange of stories between Perdita and her mother—one that certainly promises to tell us realities and truths not otherwise available—by asserting "[t]here's time enough for that"; and Leontes will ask Paulina to "hastily" lead them all off-stage where they can "demand and answer" questions of one another and explain the part each has "performed in this wide gap of time since first / We were dissevered" (5.3.129, 154-156). So while the energy of the final act seems to be leading toward a satisfying completion for the audience—a witnessing of trauma's therapeutic release into narrative—key members of the on-stage audience tasked with listening to the impossible story of her departure ultimately run from the challenge. Leontes in particular would rather focus on the separations and temporal gaps that relate to him, and to hear accounts of his own choosing.

But it is Perdita, "hearing of her mother's statue," who has requested this audience— and appropriately so (5.2.85). She, like her lost brother, shares a unique part of her mother's traumatic history and the grief at its core. Only, unlike the fatal spread of Mamillius's pas-

sions to Hermione, the passage of grief between the newborn Perdita and her mother moves from adult to child and is figured as therapeutic: according to the midwife, Hermione "receives much comfort in't [her baby]; says, 'My poor prisoner, I am innocent as you'" (2.2.31-32). Hermione also has nursed her daughter (but not her son), a physical commingling that embodies and enables their shared history, as Hermione tells it: Perdita's "innocent mouth" was full of her mother's "innocent milk" when she was "haled out to murder" by her father (3.2.98-99).[42]

Perhaps it is only in unconsciousness (or pre-consciousness) that one can listen to trauma's impossibility. Perdita's early imbibing of her mother's lamentable story seems to have made her open to more of it—targeting her as the one to be wounded by attentiveness to her father's account of Hermione's demise; and, finally, to be chosen to hear it from her mother herself, for it is to Perdita that Hermione directs her speech once she revives.

Furthermore, Perdita is the play's most palpable expression of traumatic loss, in name and action: she endures two unwilled departures at once, both of them untimely and violent (one from her mother and one from her homeland); and she is denied all access to this part of her story. When Hermione is finally given center stage to tell hers, it is no wonder she speaks exclusively to the one person present who is experienced in carrying this kind of unknowable, impossible history:

> Tell me, mine own,
> Where hast thou been preserved? Where lived? How found
> Thy father's court? For thou shalt hear that I,
> Knowing by Paulina that the oracle
> Gave hope thou wast in being, have preserved
> Myself to see the issue. (5.3.124-129)

In this, her one and only speech after sixteen years, Hermione continues to exacerbate the dramatic inconsistencies that have marked her repeated, belated returns in the play. One obvious disjunction is that Hermione was present when the oracle was read in act 3, yet she seems to rely on Paulina's knowledge of it. This is a classic feature of a traumatic response: Hermione cannot remember the episode because it involved the loss of her son—an event that she could not assimilate fully at the time, and so one to which she cannot return. Like her ghost, she is incapable of articulating his loss and its painful effects on her here.

But, like that spectral time-traveler, Hermione-as-living-statue also refigures the victim's departure from the site of trauma and her protection through cognitive detachment. Hermione's sixteen-year retreat from the world enacts a more peaceful version of the insane insensibility Constance imagines when she hopes to forget her grief and numb herself to its infectious takeover of bodies and spaces. Hermione will not need to go mad in order to manage grief's pain, and she will not die in a frenzy. In Shakespeare's late return to trauma's creative possibilities, he invents a final embodied image that is profoundly self-affirming—one that transforms the traumatized individual's unwilled departure into a conscious choice for conservation: "[I] have preserved / Myself to see the issue."

It may be impossible for Hermione to see the issue from which she departed in act 3—the lost son whose fatal fears were caught and absorbed by his mother, hastening her own speedy exit. But this last image of self-preservation allows her to connect with and assimilate the loss of her other issue—the one who, like her, carries a traumatic history, yet "has been preserved"— and to offer the possibility of listening to her story in return, even if we will not be privy to it: "Tell me, mine own."

**Funding:** This research received no external funding.

**Institutional Review Board Statement:** Not applicable.

**Informed Consent Statement:** Not applicable.

**Data Availability Statement:** Not applicable.

**Conflicts of Interest:** The author declares no conflict of interest.

## Notes

[1] All quotations from Shakespeare's works are from (Greenblatt 1997).

[2] Foundational studies include: (Paster et al. 2004; Paster 2004; Floyd-Wilson and Sullivan 2007; Johnson et al. 2014).

[3] The list includes more male characters than female, which suggests that early moderns did not necessarily consider women to be more prone to dying of grief: see, for example, King Lear, Enobarbus, and Brabantio.

[4] Sullivan notes that deaths by grief were not likely to be understood as suicides, since there were different categories to cover this type of death (Sullivan 2013). See, as well, Clodagh Tait's study of depositions from the 1641-42 Irish rebellion, in which she demonstrates that "the emotion-word most often used by deponents is grief," and that "death might ensue when grief became excessive" (Tait 2017, pp. 271–72).

[5] Darryl Chalk and Mary Floyd-Wilson unite a recent collection of essays around the emerging field of what they term "early modern contagion theory." The volume explores how early modern writers focused on the "possibility of contagious transmission, the idea that someone might be infected or transformed by the presence of others, through various kinds of exchange"—emotional, intellectual, and environmental (Chalk and Floyd-Wilson 2019, 11 and 1). See, as well, Eric Langley's *Shakespeare's Contagious Sympathies* (Langley 2018).

[6] (Bradwell 1636, pp. 34–37). The full title of Bradwell's work is *Physick for the Sicknesse, Commonly Called the Plague.*

[7] Caruth outlines these phenomena in the introduction to her edited collection *Trauma: Explorations in Memory* (Caruth 1995, p. 4).

[8] In their recent contribution to early modern emotion studies, Richard Meek and Erin Sullivan argue that previous scholars have given too much weight to Galenic medicine and humoral theory, and that "we need to give more attention to the other systems of knowledge and representation that people used to conceptualise and articulate emotional experience." They note that "early modern theories of mind, soul and will overlapped with those of the body in complex and often contested ways, destabilizing any straightforward explanation of how emotional experience might be produced" (Meek and Sullivan 2015, p. 6). See, as well, Helen Hackett's argument that early modern doctrines based in Stoicism, Platonism, and Christianity in particular promoted the detachment of the mind from the body (Hackett 2022, pp. 47–76).

[9] A Messenger delivers the story to King John: "And as I hear, / The Lady Constance in a frenzy died / Three days before; but this from rumour's tongue / I idly heard; if true or false I know not" (4.2.121-124).

[10] Caruth elaborates on this concept in her introduction to *Trauma: Explorations in Memory* (Caruth 1995, pp. 4–5).

[11] (Peters and Richards 2021, pp. 2, 11). Peters and Richards' introduction provides a thorough overview of the scholarship on and debates surrounding early modern trauma and emotions over the past twenty-five years. See, as well, Lisa Starks-Estes' introduction to her *Violence, Trauma and 'Virtus' in Shakespeare's Roman Poems and Plays* (Starks-Estes 2014, pp. 21–29); and Patricia A. Cahill's study, in which she argues that "while no literal lexicon of trauma exists in the early modern period, one can discern in the period's war plays what contemporary theorists have described as the repetitive structure characteristic of trauma" (Cahill 2008, p. 8).

[12] Pollmann makes this argument in her book *Memory in Early Modern Europe, 1500–1800* (Pollmann 2017, pp. 184–85).

[13] Tanya Pollard connects Hermione's return to Shakespeare's rewriting of Euripedes' *Alcestis*, arguing that the play "deepens Shakespeare's longstanding engagement with the ghosts of Greek tragic women by turning concertedly to the power of maternal passions" (Pollard 2017, p. 188). Felicity Dunworth discusses the legacy of the Griselda story and argues that "Hermione's pregnant body" is transformed "to a signifier of sacrifice and suffering as her young son Mamillius is removed from her and from the stage" (Dunworth 2010, p. 207). For exemplary readings of Hermione's connections to the Virgin Mary and Catholicism more broadly, see (Vanita 2000; Dolan 2007).

[14] For a sampling of approaches to Hermione's pregnant and nursing body, see (Adelman 1992, pp. 219–238; Krier 2001, pp. 234–248; Bicks 2003, pp. 22–59; Ephraim 2007).

[15] See, for example, Maureen Quilligan's contribution to *The Oxford Handbook of Shakespeare and Embodiment* (Quilligan 2016).

[16] Caruth elaborates on this idea of the therapeutic listener in her introduction to *Trauma: Explorations in Memory* (Caruth 1995, p. 10).

[17] (Hirschfeld 2003, pp. 439–440). Deborah Willis' article on *Titus Andronicus* was another important early application of trauma theory to Shakespeare. Willis uses trauma theory to explore how Tamora's initial maternal grief transforms into revenge, which, she argues, can provide "an emotional container for trauma" (Willis 2002, p. 37). See, as well, Thomas P. Anderson's work on early modern trauma and performance, which he frames within the religio-political context of the Reformation: "the belated appearance of the past transforms the present with its insistent return," he argues, and these returns disrupt artistic mediations of the past (Anderson 2006, p. 6).

[18] Michael Bristol argues that Hermione's connection to "reproductive time" relegates her story to the play's "margins, entailments, and structuring absences," where it is "systematically and violently excluded from the social time and space represented in this play" (Bristol 1996, p. 174). While I would not wish to reduce her solely to her maternal associations, I find his articulation of these two different kinds of time helpful for thinking about the problems of telling and accommodating traumatic history within conventional spatio-temporal structures.

19 (Dawson 2021, p. 242). Her work complements Lisa Starks-Estes' conviction that "literature and other arts offer human beings a vehicle through which they can refigure and reconstitute traumatic experience, in an effort to explore the tenuous boundary between the internal and the external, the subject and the event, the past and the present" (Starks-Estes 2014, p. 32). See, as well, Catherine Silverstone's study of "the ongoing and pernicious effects of various forms of violence as they have emerged in contemporary performances of Shakespeare's texts" (Silverstone 2011, p. 2).

20 See, for example, (Wood 2002, pp. 185–213; and Paster 2004, pp. 71–72).

21 In his contribution to *Embodied Cognition and Shakespeare's Theatre*, Michael Schoenfeldt observes: "It is perhaps no accident that the essays which hold the central position in this volume dedicated to the relations between body and mind focus on *The Winter's Tale*. In the famously unprovoked sudden-onset jealousy that erupts out of nowhere, physiology is cognition, body is mind" (Schoenfeldt 2014, p. 106). The three essays to which he refers indeed all focus on Leontes' jealousy.

22 Donovan Sherman, for example, considers how the disappearance of Mamillius represents "the unknowable traumatic event" at the center of the play with which Leontes must grapple. Like many critics before him, he identifies Hermione's healing of her husband as her most significant act: "If Mamillius is the failed connection of theatrical traumatic event to the textual narrative, then Hermione-as-statue is the successful implementation of this coherence. She bridges the gap before his [Leontes'] eyes" (Sherman 2009, p. 210). In a more recent reading, Paula Marantz Cohen identifies Leontes as the instigator rather than the victim of the play's traumatic events, but still interprets Hermione as the means through which his redemption is enacted. (Cohen 2021, p. 138). See, as well, Sarah Beckwith's argument that "[i]f Shakespearean ghosts have been concerned with forgetting, the new paradigm articulated in *The Winter's Tale* is concerned with recollection, re-imagined through the paradigm of repentance and resurrection" (Beckwith 2011, p. 128).

23 See, for example, Edward Reynolds, who quotes Ulysses in his treatise on the passions: "Had I foreseene this Griefe, or could but feare it, I then should have compos'd my selfe to beare it" (Reynolds 1640, p. 224).

24 Sullivan's essay is part of a collection of work by scholars on early modern emotions (Sullivan 2015, p. 32).

25 The quotation appears in Reynolds' *A Treatise of the Passions and Faculties of the Soule of Man* (Reynolds 1640, pp. 54–55).

26 Wright's *Passions of the Mind in Generall* went through several early modern editions (Wright 1604, p. 32).

27 The midwife Jane Sharp writes that a pregnant woman should "avoid violent passions, as care, and anger, joy, fear, or whatsoever may too much stir the blood" if she wishes to prevent "abortment" and bring her child to term (Sharp 1671, p. 224).

28 Long's essay is part of an edited collection on violence and trauma in British theater (Long 2009).

29 In her work on mentally distressed war veterans and their claims of traumatic injury in early modern judicial proceedings, Ismini Pells tracks the vocabulary of grief and sorrow and argues that "traumatic language was regarded as helpful to a petition's success, indicating a broader contemporary sympathy for psychological wounds" (Pells 2021, p. 136). While Leontes is not a petitioner by any means, in having him articulate the physical nature of his grief here, Shakespeare may be pointing to the ways in which traumatic language could be manipulated in the hopes of arousing sympathy in an audience.

30 (Peterson 2010, pp. 145–46). In her chapter on the strangling of the womb, the midwife Jane Sharp describes the appearance of death that it can cause and attributes this not just to the uterus, but to a "sudden fright," or "a bad news related" (Sharp 1671, p. 321).

31 (Wright 1604, p. 71). Wright's image of man's tempest-tossed emotions is frequently cited by early modern scholars.

32 (Greene 1588, p. 22). Shakespeare based his plot on Greene's sixteenth-century prose romance.

33 Richard Wilson describes Hermione, in what he calls this "bizarre dream sequence," as a "classic instance of female abjection at the turning-point of the play," the repellant figure of the 'undead' woman returning from the morgue, and a "vampiric spectre" (Wilson 2014, p. 205). In her insightful treatment of Hermione's ghost, Frances Dolan connects it to the specter of Catholicism, but still considers its appearance to be "a dream" (Dolan 2007, p. 225). Katherine Kellett reads Hermione's ghost as an engagement with the 1590s genre of complaint poems voiced by females returning from the grave. Like Dolan, she describes the ghost as part of "Antigonus' dream" (Kellett 2013, p. 25). Stephen Orgel is the rare critic to argue that Antigonus is convinced it "was an apparition, not a dream" (Orgel 1996, p. 153fn).

34 Shakespeare gives Bohemia a shoreline, although it was and is a landlocked country (now known as the Czech Republic). Andrew Gurr was one of the first critics to view this factual slip (like Shakespeare's decision to swap Greene's settings of Bohemia and Sicilia) as an artful choice, arguing that he did so "to flout geographical realism, and to underline the reality of place in the play" (Gurr 1983, p. 422). This disruption of geographical space makes perfect sense within the traumatic framework I am establishing here.

35 (Caruth 1996, p. 4, emphasis mine). Caruth describes this connection between trauma and the Greek word for *wound* in her book, *Unclaimed Experience: Trauma, Narrative, and History*.

36 Caruth discusses this witnessing of impossibility in her collection, *Trauma: Explorations in Memory* (Caruth 1995, p. 10).

37 Caruth is referring here to Terr's work on remembered images and trauma (Caruth 1995, p. 10).

38 Jeanette Winterson entitles her novelized retelling of the play *The Gap of Time*, and uses this image in her memoir as well to address the traumatic experience of adoption—of always seeking, but never finding one's origins, and always measuring love by loss. Stories, she writes, are a way to compensate for these gaps by creating a new, possibly therapeutic space: "When we tell a

story we exercise control, but in such a way as to leave a gap, an opening. . . . And perhaps we hope that the silences will be heard by someone else, and the story can continue, can be retold" (Winterson 2011, p. 8).

39     See Roger Luckhurst's *The Trauma Question* (Luckhurst 2008, p. 80).

40     In his influential study, Stanley Cavell argues that Hermione "*is* the play," and that the audience is "her, and the play's, issue." (Cavell 2003, p. 219). More recently, Michael Witmore argues that Hermione's actions, "lawful as they are said to be, are always contained within the realm of art, even if we learn that she has continued her very real life in order to see the oracle fulfilled." She is "an allegory for the vivifying power of art" (Witmore 2007, p. 164).

41     J.R. Bernard considers how these reactions testify "to the impossibility of properly staging the difficult conversations that undoubtedly follow." His reading of trauma in the play does not consider Hermione's experience of it, although he offers a moving account of Mamillius' loss: "the play's wondrous final act looks elsewhere, but Mamilius' [sic] sad tale is not that easily forgotten" (Bernard 2018, pp. 196–97). Some modern productions dramatize this painful lingering by bringing the specter of Mamillius on stage at the end. In the final moment of Slobodan Unkovski's 2000 production for the American Repertory Theatre, for example, the boy's hand appears and slides along a wall, visible only to Hermione, before it disappears and the lights go out.

42     Before ordering Mamillius' removal from Hermione, Leontes tells her he is "glad you did not nurse him" (2.1.58). See Donna C. Woodford's analysis of nursing in *The Winter's Tale* (Woodford 2016).

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

Wright, Thomas. 1604. *The Passions of the Mind in Generall*. London.