# Peer review of "Fear of the Queen’s Speed: Trauma and Departure in The Winter’s Tale"

_humanities, doi:10.3390/h11060156_

Round 1

Reviewer 1 Report

This is an interesting and scholarly article which makes more sense of Antigonus' vision of a dead Hermione than anything I have read previously.  It certainly deserves to be published, but there are some minor things I'd like to see addressed.  The opening sentence assures me that everyone knows what 'the humoral body-mind' is.  Actually that phrase is not familiar to me and while I can certainly guess at what it might mean, I don't know what precisely it does mean.  Further down on the same page I am also left unsure whether the author really means 'immanent death' or whether it should be 'imminent'.  I'd note too that while referring to 'now in 2022' has an obvious attraction given the pandemic, it will date very quickly (in less than three months indeed).  I'm also not sure about Hermione defying the laws of time, space and motion more than any other character in the canon; how about Ariel - indeed how about thinking of Hermione as a quasi-magical being in something of the same way as Ariel?  Especially given that he gets imprisoned in a tree and she is 'imprisoned' as a statue.  Finally early moderns' lived experiences not early moderns lived experiences - I know Microsoft Word was programmed by someone who didn't understand this, but nevertheless a plural possessive needs an apostrophe after the s.

Author Response

Thank you for these comments.

1) I have defined body-mind now at the start of the essay and also removed most references to it throughout the essay in order to present a more nuanced reading of how trauma affects the mind separately from the body.

2) I have added "human" to the description of Hermione when I claim that she "defies the laws of time, space, and motion to an extent unmatched by any other human character in his canon," so as to address the reader's comment that Ariel does so as well. I do not with to read Hermione as semi-magical.

3) I have corrected the typos and other errors in style and spelling.

Reviewer 2 Report

This is a very well researched and written article. It makes clear and effective points about the relation of trauma theory to The Winter's Tale, and it is well worth publishing.

My only caveats stem from the fact that there are more than just humoral approaches to early modern emotions. Erin Sullivan, for example, some of whose work is cited here, makes the case for more rhetorical contexts for understanding the emotions in the period in her book Beyond Melancholy, and she is just one of a number of recent critics (Mary Morrissey, Kurt Essary, to name just two others), pointing out the many ways in which emotions could be understood at the time. Helen Hackett lays these ways out very clearly in her recent book The Elizabethan Mind.

So the article's sole focus on the humoral emotions - and its repeated use of 'body-mind' - risks simplifying the context of emotions in the period. I also find the whole 'body-mind' phrase problematic when used in the context of theatrical representation, since any play is, after all, dealing with characters and not actual bodies in the way the phrase implies.

I would therefore recommend - not require - that the author take into account the varying views on early modern emotions a little more fully, and perhaps rein in the repetition of 'body-mind' in the article, as it is not only wearying but also inaccurate, in my view.

However, overall this is a very good article and well worth publishing. These points are mentioned only to deepen the article's contextualisation of emotions and to enhance its ability to draw in those who might be less convinced of the validity of the monist view of the body and the emotions.

Author Response

I am grateful to this reviewer for calling attention to my over-reliance on the term and concept "body-mind" and on a  humoral view of the emotions. I have substantially revised the essay removing most references to "body-mind" and (as per their recommendation) explicitly noted how other early modern discourses separated the mind from the body. This change has, I believe, improved my argument and allowed me to present a more nuanced reading of how trauma can affect the mind separately from the body.